# Prevalence and factors associated with anemia among women of reproductive age in seven South and Southeast Asian countries: Evidence from nationally representative surveys

**Dev Ram Sunuwar**[1,2]*, **Devendra Raj Singh**[2,3], **Narendra Kumar Chaudhary**[4], **Pranil Man Singh Pradhan**[5], **Pushpa Rai**[6], **Kalpana Tiwari**[7]

1 Department of Nutrition and Dietetics, Armed Police Force Hospital, Kathmandu, Nepal, 2 Department of Public Health, Asian College for Advance Studies, Purbanchal University, Lalitpur, Nepal, 3 Research and Innovation Section, Southeast Asia Development Actions Network (SADAN), Lalitpur, Nepal, 4 Department of Radiology, Nepal Orthopaedic Hospital, Kathmandu, Nepal, 5 Department of Community Medicine, Maharajganj Medical Campus, Institute of Medicine, Tribhuvan University, Kathmandu, Nepal, 6 Department of Emergency, Patan Academy of Health Sciences, Lalitpur, Nepal, 7 Department of Nutrition and Dietetics, College of Applied Food and Dairy Technology, Purbanchal University, Kathmandu, Nepal

* devramsunuwar@gmail.com

**Data Availability Statement:** We used data of Demographic and Health Surveys that are publicly

## Abstract

### Background

Anemia remains a major public health challenge with high prevalence among women in South and Southeast Asian countries. Reductions in anemia rates have been stalled, despite the implementation of different maternal health and nutrition programs. This study aimed to assess the prevalence and factors associated with anemia among women of reproductive age in seven selected South and Southeast Asian countries.

### Methods

This cross-sectional analysis utilized data from the most recent demographic and health surveys from seven selected South and Southeast Asian countries (Bangladesh, Cambodia, India, Maldives, Myanmar, Nepal, and Timor-Leste) between 2011 and 2016. This study included 726,164 women of reproductive age. Multiple logistic regression was performed to assess the factors associated with anemia among women for each country separately.

### Results

The combined prevalence of anemia was 52.5%, ranged from 22.7% in Timor-Leste to 63% in the Maldives. Results from multiple logistic regression suggest that likelihood of anemia is significantly higher among younger women (15–24 years), women with primary or no education, women from the poorest wealth quintile, women without toilet facilities and improved water sources, underweight women, and women with more than one children born in last five years in most of the countries.

available and can be freely downloaded upon the formal request from the DHS website (https://www.dhsprogram.com/data/available-datasets.cfm)

**Funding:** The author(s) received no specific funding for this work

**Competing interests:** The authors have declared that no competing interests exist

## Conclusions

The prevalence of anemia is high among women of reproductive age in the seven selected South and Southeast Asian countries. The results of this study suggest that various household, environmental and individual factors contribute to the increased likelihood of anemia. Evidence-based, multidisciplinary policies and programs targeting mothers' health and nutrition status, in addition to scaling-up women's education and socioeconomic status, are warranted to combat anemia.

## Introduction

Anemia, which is defined as a hemoglobin (Hb) concentration less than normal [1], remains a major public health challenge with a prevalence rate of 47% among non-pregnant and 52% in pregnant women in South and Southeast Asian (SSEA) countries [1,2]. Globally, approximately one-third of the population is affected by anemia and its epidemiology varies according to population age, sex, socio-cultural contexts, and geographical regions [2,3]. Women of reproductive age (WRA) are physiologically more prone to anemia due to persistent menstrual blood loss and the demands of repeated childbearing and pregnancy [4]. According to the World Health Organization (WHO), worldwide approximately 39% of WRA, 46% of pregnant women were affected by anemia in 2016 [5]. Anemia is associated with increased morbidity and mortality in women, child growth faltering, impairment of cognitive function, increased chances of various kinds of infection, loss of productivity from impaired work capacity resulting in substantial economic burden to the family and entire population [6–8].

The prevalence of anemia varies according to geographic regions. Sub-Saharan Africa (SSA) and South Asia had the highest prevalence of anemia across all age groups [9]. Likewise, at the country level, anemia among WRA is a moderate-to-severe public health problem (20% or greater as defined by WHO) in the majority of the developing countries [2,10]. Determinants and distribution of prevalence of anemia in a population include a complex interplay of political, ecological, social, and biological factors [4]. In most of the countries, anemia varies by socioeconomic factors such as education, household wealth status, occupation, and residence [2,6]. A pooled analysis conducted by Balarajan et al [6] reported that the risk of anemia among women living in the lowest wealth quintile, with no education, and also differed by urban or rural settings. Likewise, previous studies have highlighted the probable causes of anemia among women including undernutrition, repeated childbearing, pregnant and lactation, inadequate dietary intake during pregnancy, inadequate water hygiene and sanitation status, rural residency, and parasitic infection [11–13]. Though several causes are associated with anemia, iron deficiency anemia is the most common type of anemia worldwide that is usually caused by inadequate intake of iron-rich foods in regular diets and excessive loss of red blood cells or a combination of both [3]. Various studies conducted from the developing countries have reported a high prevalence of iron deficiency anemia among pregnant women [14,15]. In low and middle-income countries (LMICs), the causes of anemia can be broadly classified into three major groups: nutritional deficiencies, infectious diseases, and genetic hemoglobin disorders [16]. Anemic condition in mothers has multiple adverse health consequences, such as the increased risk of miscarriage, stillbirth, premature, and low birth weight [17–19]. Approximately 20% of maternal deaths are caused by anemia and it is also considered as an additional risk factor for 50% of all maternal deaths [15,20]. To prevent anemia among WRA, pregnant

women, and young children, different approaches at population and individual levels have been implemented [21]. For example, micronutrient supplementation among adolescents girls and pregnant women, food fortification, provision of nutrition education, counseling, and iron-rich food-based diet plan to at-risk populations are strategies used to improve dietary diversity and quality [21,22].

Despite the noticeable progress in socio-economic and health status in the majority of low-income countries in the SSEA region, countries are still challenged with reducing a high burden of malnutrition among WRA [21]. The prevalence of anemia has declined from an estimated 40% in 1990 to an estimated 33% in 2016 with a decrease of roughly seven percentage points over this time [3]. However, the progress made in reducing the prevalence of anemia is far less than the expected and its socioeconomic burden, particularly in resource-poor countries, is still a major concern [3]. The prevalence of anemia among non-pregnant women in South Asian countries declined slightly from an estimated 53% in 1995 to an estimated 47% in 2011, whilst the prevalence of anemia among pregnant women in the same region was almost stagnant (53% in 1995 and 52% in 2011) [2]. The Southeast Asian region has little progress in the reduction of anemia among WRA [3]. Considering the slow progress in the reduction of anemia prevalence, during the 2012 World Health Assembly, the World Health Organization endorsed a target of a 50% reduction of anemia among women of reproductive age by 2025 [17,23]. Based on the global prevalence of 29–38% anemia among WRA in both pregnant and non-pregnant women in 2011, a reduction of 1.8–2.4% points per year would be required to meet this target [9]. The Sustainable Development Goals (SDGs) also outline targets under SDG-2 to reduce the different forms of malnutrition among children under five years of age, pregnant women, lactating mothers, adolescents girls, and older people [24].

To achieve the WHO global nutrition targets 2025 and nutrition targets of the Sustainable Development Goals-2030, it is necessary to generate adequate evidence on contextual determinants of anemia to contribute to the development of timely interventions in anemia prevention. In addition, exploring the commonalities and differences across the states in the South and Southeast Asia region can inform the regional policy. Although previous studies have attempted to estimate the prevalence of anemia, very few studies have utilized nationally representative data to investigate the prevalence and determinants of anemia among WRA in the SSEA region. Furthermore, the updated evidence on prevalence and factors associated with anemia among at-risk populations is lacking for the SSEA context.

Therefore, this study aims to identify the prevalence and factors associated with anemia among WRA in seven selected SSEA countries.

## Methods

### Data sources

This study utilized the data from the Demographic and Health Surveys (DHSs) of seven selected SSEA countries conducted between 2011 and 2016 (Bangladesh DHS 2011, Cambodia DHS 2014, India NFHS 2016, Maldives DHS 2016, Myanmar DHS 2015, Nepal DHS 2016, and Timor-Leste 2015). We excluded the remaining SSEA countries due to a lack of availability of data on anemia among WRA (15–49 years old). All WRA women from the selected countries were included in the analysis. DHS is large, nationally representative, household surveys which are usually conducted by in-country/local institutions and are funded by the United States Agency for International Development (USAID), with technical assistance from ORC (Opinion Research Corporation) Macro international Inc. Calverton, Maryland, USA [25].

The DHS datasets for the selected countries were obtained from the DHS program website (URL: https://www.dhsprogram.com/data/available-datasets.cfm) after receiving approval to

access and download the DHS data file. All DHS utilizes a multistage cluster sampling design to provide representative estimates for all enumeration areas. Probability proportional to size (PPS) methodologies areas employed by DHS to include both rural and urban residence, followed by a random selection of households from within the selected clusters or enumeration areas. All surveys included sample weights in the dataset. A more detailed sampling methodology of DHS has been published elsewhere [26–32]. Trained interviewers collected data on socio-demographic factors, health, and nutrition status. All eligible women were interviewed by trained interviewers in a local language.

## Analytic sample

WRA (pregnant and non-pregnant) with available data on anemia, complete socio-demographic, and nutritional information were included in the analysis. A total sample used in the analysis for each survey is given in S1 Table.

## Outcome variables

According to the WHO, for non-pregnant and pregnant women aged 15–49 years, any form of anemia was defined as hemoglobin concentration <12 g/dL, and <11 g/dL respectively [1]. Hemoglobin level in capillary blood was assessed using HemoCue rapid testing technique in all seven SSEA countries. For further analysis of the outcome variable, the categories of anemia were further dichotomized into 'anemic' and 'not anemic'.

## Predictor's variables

Predictors of anemia were selected based on the literature review to the risk of development of anemia among women in LMICs including the South and Southeast regions [2,4,6,16].

The variables used in this study were similar across the seven SSEA countries. The predictor' variables used in this study included household, environmental and individual factors. Household factors include residence, education level, wealth status, marital status, occupation, type of toilet facility use, and water source. The place of residences of the respondents was classified as rural and urban. The educational status was classified according to the DHS categories. The wealth index was already included in all datasets and further classified as poor middle and rich. The wealth index is a standardized measure, quantified using principal component analysis (PCA) from household assets [33]. The wealth index variables were already included in the DHS dataset as five quintiles ranked as poorest, poorer, middle, richer, and richest. In this study, we further re-categorized as poor (poorest and poorer), middle and rich (richer and richest) for analysis [13]. Marital status was classified as being married/living together, never in the union, and widowed/divorced/separated. The respondent's occupation was dichotomized as working and not working. Source of drinking water and type of toilet facility used were classified into improved and not improved. Similarly, individual factors include age, BMI, and iron supplementation during most recent pregnancy, birth spacing, and ever terminated pregnancy (Whether the respondent ever had a pregnancy that terminated in a miscarriage, abortion, and stillbirth), antenatal checkup (ANC) during pregnancy (number of antenatal visits during pregnancy), currently breastfeeding, and total children ever born. The age of the respondents was recoded into 15–24, 25–34, and 35–49 years. Body mass index (BMI) was categorized into underweight ($<18.5$ kg/m$^2$), normal (18.5–24.9 kg/m$^2$), overweight (25–29.9 kg/m$^2$) and obesity ($\geq$30 kg/m$^2$) according to the WHO classification. Iron supplementation during most recent pregnancy, ever terminated pregnancy, and currently, breastfeeding was classified as no and yes. Birth spacing was categorized into none, one and

more than one year. Antenatal checkup visits during the pregnancy were classified as no, less than five times, and more than or equals to five times visit.

## Data analysis

Data were analyzed using Stata/MP version 14.1 (StataCorp LP, College Station, Texas). The 'svy' command was used to adjust for enumeration areas (EAs) and disproportionate sampling weight and non-response. The datasets from seven countries were pooled and merged for analysis. For each country, we estimated overall and country-level weighted prevalence rates and their 95% confidence intervals (CIs) of anemia among WRA. The weighted prevalence of anemia among WRA was examined according to household, and individual factors. Multiple logistic regression models were performed to assess the association between anemia and associated factors in each country. To prevent statistical bias in the multiple logistic regression model, we examined and reported multicollinearity among the predictor variables using variation inflation factors (VIF). In this study, we used "10" as a cut-off value for the maximum level of VIF [34]. Results were presented as adjusted odds ratio (aOR) with 95% confidence intervals (CIs). P-values <0.05 were considered as statistically significant. Additionally, ArcGIS software version 10.8 was used to generate the study area map, and base files of the administrative national boundaries for seven selected SSEA countries were obtained from Natural Earth [35].

## Ethical consideration

The study protocols were reviewed and approved by the National Ethics Committee of the Bangladesh Medical Research Council, Cambodian National Ethical Committee for Health Research, Ethics Review Board of the International Institute for Population Sciences, Mumbai, India, Ethics Review Committee on Medical Research, Ministry of Health, Maldives, Ethics Review Committee on Medical Research, Ministry of Health and Sports, Myanmar, Nepal Health Research Council (NHRC), Timor-Leste Ministry of Health, [26–32] and the Institutional Review Board (IRB) of ICF Macro. We registered and requested for access to data from the DHS website (URL: https://www.dhsprogram.com/data/available-datasets.cfm) and received an approval to access and download the DHS data file. DHS programs collect data following written informed consent from each individual. This study did not include any identifying of individuals during the variables selection process.

## Results

A total of 726,164 women aged 15–49 years were included in the pooled analysis of seven SSEA countries (Table 1).

### Prevalence of anemia among women of reproductive age

The overall weighted prevalence of anemia among these populations was 52.5% ranged from 22.7% in Timor-Leste to 63% in the Maldives (Fig 1). The prevalence of mild, moderate and severe anemia was 39.4%, 12.1%, and 1% respectively in this region. Whereas the prevalence of mild and moderate anemia was higher in the Maldives (mild anemia: 49.1% and moderate anemia: 13.3%) and lower in Timor-Leste (mild anemia: 18.5% and moderate anemia: 3.7%), while the prevalence of severe anemia was less than 1% in most of the countries except India (1%) (Fig 2 and Table 1).

The prevalence of anemia among WRA in pooled analysis was higher in rural areas compared to urban areas in most of the countries (ranged from 42.6% in Nepal and 54.2% in India), except Maldives (urban Vs rural: 73.3% Vs 55.6%) and Timor-Leste (urban Vs rural:

**Table 1. Socio-demographic information with weighted prevalence rates for anemia among women of reproductive age in seven South and Southeast Asian countries.**

| | Anemia status % (95% CI) | | | | | | | |
|---|---|---|---|---|---|---|---|---|
| | Cambodia (DHS 2014) | India (NFHS 2016) | Maldives (DHS 2016/17) | Myanmar (DHS 2015) | Nepal (DHS 2016) | Timor-Leste (DHS 2016) | Bangladesh (DHS 2011) | SSEA % |
| N | 11,286 | 679,445 | 6,653 | 12,489 | 6,414 | 4,201 | 5,676 | 726,164 |
| **Overall prevalence** | 45.3 [44.1, 46.5] | 53.1 [52.8, 53.3] | 63 [61.1, 64.9] | 46.5 [45.1, 47.9] | 40.7 [38.5, 42.9] | 22.7 [21.1, 24.3] | 42.4 [40.6, 44.2] | 52.5 [52.3, 52.8] |
| **Study variables** | | | | | | | | |
| **household factors** | | | | | | | | |
| **Residence** | | | | | | | | |
| Urban | 39.3 [37.4, 41.3] | 50.8 [50.3, 51.3] | 73.3 [69.3, 76.9] | 46.4 [43.3, 49.5] | 39.6 [36.8, 42.4] | 24.7 [21.8, 28] | 36.1 [32.5, 39.7] | 50.5 [50, 50.9] |
| Rural | 46.7 [45.3, 48.1] | 54.25 [54, 54.5] | 55.6 [53.8, 57.3] | 46.5 [44.9, 48] | 42.6 [39, 46.3] | 21.7 [19.8, 23.6] | 44.6 [42.5, 46.7] | 53.6 [53.4, 53.8] |
| **Education** | | | | | | | | |
| No education | 47.7 [44.4, 51.1] | 56.4 [56.1, 56.7] | 64.2 [55.9, 71.7] | 44.9 [41.8, 48.1] | 41.5 [38.2, 44.9] | 24.6 [21.7, 27.6] | 47 [43.8, 50.2] | 55.8 [55.5, 56.2] |
| Primary | 47.5 [45.8, 49.2] | 54.6 [54.1, 55] | 62.2 [59.2, 65.1] | 47.2 [45.3, 49.1] | 38.4 [34.2, 42.8] | 23.8 [20.1, 28] | 44.6 [41.8, 47.5] | 53 [52.9, 53.9] |
| Secondary | 42.6 [40.8, 44.3] | 52.1 [51.8, 52.4] | 60.7 [58.2, 63.1] | 46 [43.9, 48.1] | 42.6 [39.6, 45.7] | 21.7 [19.7, 23.9] | 39.1 [36.5, 41.6] | 51.7 [51.4, 52.1] |
| Higher | 37.5 [33.3, 41.8] | 47 [46.9, 48.2] | 69.6 [65.4, 73.5] | 47.2 [43.6, 50.7] | 36.7 [32.3, 41.3] | 21.6 [17, 27.1] | 42.4 [40.6, 44.2] | 47.5 [46.9, 48.1] |
| **Wealth status** | | | | | | | | |
| Poor | 1.1 [48.9, 53.1] | 56.8 [56.5, 57.1] | 56.7 [54.5, 58.9] | 47.5 [45.5, 49.5] | 37.1 [34.5, 39.9] | 21.8 [19.3, 24.6] | 48.9 [46.2, 51.6] | 56.1 [55.8, 56.4] |
| Middle | 44.5 [41.8, 47.2] | 53.3 [52.9, 53.7] | 60.3 [56.9, 63.5] | 47.4 [44.6, 50.2] | 48.9 [45.1, 52.8] | 23.6 [19.9, 27.8] | 42.6 [39.1, 46.2] | 52.8 [52.4, 53.2] |
| Rich | 40.9 [39.2, 42.7] | 49.6 [49.1, 50] | 70.2 [66.4, 73.8] | 45.2 [43, 47.3] | 39.8 [36.4, 43.3] | 22.9 [20.4, 25.6] | 36.4 [33.7, 39.1] | 49.2 [48.8, 49.5] |
| **Marital status** | | | | | | | | |
| Never in union | 45.8 [43.6, 48.1] | 52.4 [52.1, 52.8] | 49.3 [47.1, 51.5] | 35.1 [33.5, 36.6] | 42.1 [38.5, 45.8] | 20 [17.2, 23.1] | - | 51.9 [51.5, 52.3] |
| Married/living together | 44.7 [43.3, 46.1] | 53.1 [52.8, 53.3] | 44.6 [43.1, 46.2] | 58 [56.3, 59.7] | 40.7 [38.5, 42.9] | 23.8 [22.1, 25.7] | 42.3 [40.5, 44.1] | 52.5 [52.3, 52.8] |
| Widowed/divorced/separated | 49.4 [45.5, 53.3] | 55.9 [55.1, 56.8] | 49.4 [44.9, 54] | 6.8 [6.1, 7.7] | 31.6 [24.1, 40.1] | 33.4 [23.8, 44.7] | 43.7 [40.6, 44.2] | 55.4 [54.6, 56.3] |
| **Occupation** | | | | | | | | |
| Not working | 46.3 [43.7, 49.1] | 52.9 [52.3, 53.5] | 60.8 [58.5, 63.1] | 47.4 [44.8, 49.9] | 42.9 [39.6, 46.2] | 22.8 [20.93, 24.9] | NA | 51.8 [51.3, 52.3] |
| Working | 45.1 [43.7, 46.4] | 55 [54.1, 55.9] | 65.1 [62.5, 67.5] | 46.2 [44.7, 47.7] | 39.7 [37.2, 42.2] | 22.3 [20, 24.8] | | 50.8 [50.2, 51.5] |
| **Toilet type** | | | | | | | | |
| Improved | 42.6 [41.1, 44.1] | 50.7 [50.3, 51.1] | 62.8 [60.8, 64.7] | 45.5 [43.4, 47.6] | 37.3 [35.1, 39.6] | 21.8 [19.5, 24.4] | 33.7 [29.1, 38.7] | 50.3 [50, 50.7] |
| Not improved | 49 [47.1, 50.8] | 55.8 [55.5, 56.1] | 67.7 [56.2, 77.4] | 47.1 [45.3, 49] | 51.9 [47.5, 56.3] | 23.4 [20.9, 26] | 42.4 [40.1, 44.8] | 55.2 [54.9, 55.5] |
| **Water source** | | | | | | | | |
| Improved | NA | 49.4 [48.9, 50] | 69.5 [66.3, 72.6] | 47 [44.8, 49.2] | 31.2 [28.3, 34.3] | 22.2 [20.4, 24.2] | 25.9 [19.5, 33.6] | 49.1 [48.6, 49.6] |
| Not improved | | 54.2 [53.9, 54.4] | 45.4 [36.1, 55.1] | 46 [44.3, 47.7] | 44.5 [42.1, 47] | 23.5 [20.9, 26.2] | 43.5 [41.6, 45.3] | 53.8 [53.5, 54.1] |
| **Individual factors** | | | | | | | | |

*(Continued)*

**Table 1.** (Continued)

| | Anemia status % (95% CI) | | | | | | | |
|---|---|---|---|---|---|---|---|---|
| | Cambodia (DHS 2014) | India (NFHS 2016) | Maldives (DHS 2016/17) | Myanmar (DHS 2015) | Nepal (DHS 2016) | Timor-Leste (DHS 2016) | Bangladesh (DHS 2011) | SSEA % |
| **Age group (years)** | | | | | | | | |
| 15–24 | 46.6 [44.8, 48.5] | 53.8 [53.4, 54.1] | 60.7 [57.8, 63.5] | 47.5 [45.3, 49.7] | 43.5 [40.6, 46.5] | 23.7 [21, 26.7] | 41.6 [38.6, 44.6] | 53.2 [52.9, 53.6] |
| 25–34 | 42.3 [40.2, 44.5] | 52.3 [52, 52.7] | 63.6 [60.6, 66.4] | 43.4 [41.4, 45.5] | 41.3 [38.4, 44.2] | 20.1 [17.8, 22.7] | 42.6 [40.1, 45.1] | 51.8 [51.4, 52.1] |
| ≥35 | 47.1 [45, 49.1] | 52.9 [52.6, 53.3] | 64.3 [61.6, 66.9] | 48.1 [46.2, 49.9] | 36.7 [34, 39.6] | 23.7 [20.9, 26.8] | 42.9 [40.1, 45.7] | 52.5 [52.2, 52.8] |
| **BMI** | | | | | | | | |
| Normal | 46.5 [45, 48.1] | 53.1 [52.8, 53.4] | 65.8 [63.2, 68.4] | 48.3 [46.7, 49.9] | 42.6 [39.7, 45.5] | 22.3 [20.3, 24.5] | 41.7 [39.5, 43.9] | 52.6 [52.3, 52.8] |
| Underweight | 50.5 [47.3, 53.7] | 58.6 [58.2, 59] | 66.5 [60.8, 71.7] | 52.4 [49.5, 55.3] | 48.1 [44.6, 51.6] | 24.2 [21.3, 27.3] | 51.6 [48.4, 54.8] | 58.1 [57.7, 58.5] |
| Overweight | 37.6 [34.9, 40.4] | 46.9 [46.4, 47.4] | 60.1 [57.2, 62.9] | 38.7 [35.9, 41.6] | 30.6 [27.1, 34.3] | 19.3 [14.8, 24.8] | 32 [28.3, 35.9] | 46.5 [45.9, 47] |
| Obesity | 33.4 [27.1, 40.5] | 46.8 [45.9, 47.7] | 59.7 [56.2, 63.2] | 37.4 [33.2, 41.8] | 27.9 [22.2, 34.4] | 26.7 [15.3, 42.2] | 27.9 [20.6, 36.6] | 46.6 [45.8, 47.5] |
| **Iron intake** | | | | | | | | |
| No | 45.2 [43.2, 47.2] | 56.7 [56.3, 57.2] | NA | 43.8 [41.6, 45.9] | 44.4 [41.5, 47.5] | 27.2 [24.2, 30.4] | 43.9 [41.3, 46.6] | 55.6 [55.2, 56] |
| Yes | 47.4 [37.6, 57.3] | 54.7 [53.9, 55.5] | | 40.7 [34.3, 47.5] | 38.1 [28.1, 49.1] | 22.8 [19.2, 26.9] | 30.9 [19.8, 44.9] | 54.1 [53.4, 54.9] |
| **Births in last 5 years** | | | | | | | | |
| No birth | 45.3 [44, 46.7] | 51.9 [51.6, 52.2] | 63.1 [60.8, 65.4] | 47.5 [45.9, 49.2] | 39.2 [36.9, 41.5] | 20.4 [18.4, 22.6] | 41.4 [39.2, 43.7] | 51.5 [51.3, 51.8] |
| 1 | 43.8 [41.4, 46.2] | 54.7 [54.3, 55.2] | 62.2 [59.3, 65.1] | 42.4 [40.1, 44.6] | 41.6 [38.3, 44.9] | 25.6 [22.6, 28.7] | 41.6 [38.9, 44.4] | 53.8 [53.3, 54.2] |
| >1 | 51.2 [46.4, 56.1] | 59.4 [58.8, 53.3] | 64.5 [57.3, 71.1] | 50.3 [45.3, 55.2] | 52.1 [46.5, 57.6] | 26.4 [22.3, 31] | 53.5 [48.1, 58.8] | 58.7 [58.2, 59.3] |
| **Ever terminated pregnancy** | | | | | | | | |
| No | 45.5 [44.1, 46.9] | 53.1 [52.8, 53.3] | 63 [60.9, 65] | 46.2 [44.7, 47.7] | 40.9 [38.4, 43.4] | 22.4 [20.8, 24.1] | 42.1 [40.1, 44.1] | 52.5 [52.3, 52.8] |
| Yes | 44.7 [42.4, 47.1] | 53.1 [52.5, 53.6] | 63.1 [58.6, 67.3] | 49 [45.8, 42.1] | 40.1 [36.9, 43.4] | 34.9 [23.1, 49] | 43.4 [40.1, 46.8] | 52.5 [52, 53] |
| **ANC visits during pregnancy** | | | | | | | | |
| No | 48.1 [39.3, 56.9] | 58.9 [58.2, 59.7] | 81.3 [64.6, 91.2] | 43.9 [38.2, 49.8] | 47 [36.4, 57.7] | 25.3 [19.6, 32.1] | 46.2 [42.1, 50.5] | 58.1 [57.3, 58.8] |
| <5 times | 46.8 [44.1, 49.5] | 57 [56.5, 57.4] | 59.1 [44.8, 72.1] | 44.9 [41.9, 47.9] | 44.5 [41.2, 47.8] | 28.2 [23.8, 33] | 43.4 [40.1, 46.8] | 56.1 [55.6, 56.4] |
| ≥5 times | 43.1 [40, 46.2] | 53.6 [53, 54.4] | 62.5 [59.8, 65.2] | 41.9 [38.5, 45.9] | 41.8 [35.8, 48.1] | 24.6 [21.6, 27.9] | 38.9 [32.3, 45.9] | 52.9 [52.3, 53.6] |
| **Currently breastfeeding** | | | | | | | | |
| No | 44.3 [43.1, 45.6] | 52.2 [51.9, 52.4] | 63.4 [61.3, 65.4] | 46.3 [44.7, 47.8] | 39.3 [37.1, 41.6] | 22.1 [20.2, 24] | 40.5 [38.5, 42.6] | 51.7 [51.5, 51.9] |
| Yes | 51.4 [48.5, 54.3] | 57.7 [57.3, 58.1] | 60.5 [56.5, 64.3] | 47.7 [45, 50.2] | 45.8 [42.2, 49.3] | 25.5 [22.1, 29.4] | 48 [44.9, 51.1] | 57 [56.6, 57.4] |
| **Total children ever born** | | | | | | | | |

(Continued)

**Table 1.** (Continued)

| | Anemia status % (95% CI) | | | | | | | |
|---|---|---|---|---|---|---|---|---|
| | Cambodia (DHS 2014) | India (NFHS 2016) | Maldives (DHS 2016/17) | Myanmar (DHS 2015) | Nepal (DHS 2016) | Timor-Leste (DHS 2016) | Bangladesh (DHS 2011) | SSEA % |
| No child | 46.4 [44.4, 48.4] | 51.7 [51.3, 52.1] | 60.9 [58.1, 63.5] | 48.7 [46.7, 50.7] | 41.8 [38.2, 45.4] | 20.8 [18.2, 23.6] | 39.6 [34.8, 44.5] | 51.3 [50.9, 51.6] |
| 1–4 | 44 [42.4, 45.5] | 53.5 [53.2, 53.7] | 64.3 [62, 66.5] | 43.9 [42.1, 45.6] | 40 [37.7, 42.3] | 24 [21.8, 26.4] | 41.5 [39.6, 43.4] | 52.9 [52.7, 53,2] |
| >4 | 49.4 [45.8, 52.9] | 55 [54.4, 55.6] | 62.4 [57.9, 66.7] | 50.9 [47.4, 54.4] | 42.3 [37.8, 46.9] | 23.7 [20.4, 27.5] | 48.6 [44.7, 52.6] | 54.1 [53.6, 54.7] |

Percentage and frequency are weighted.

SSEA: South and Southeast Asia.

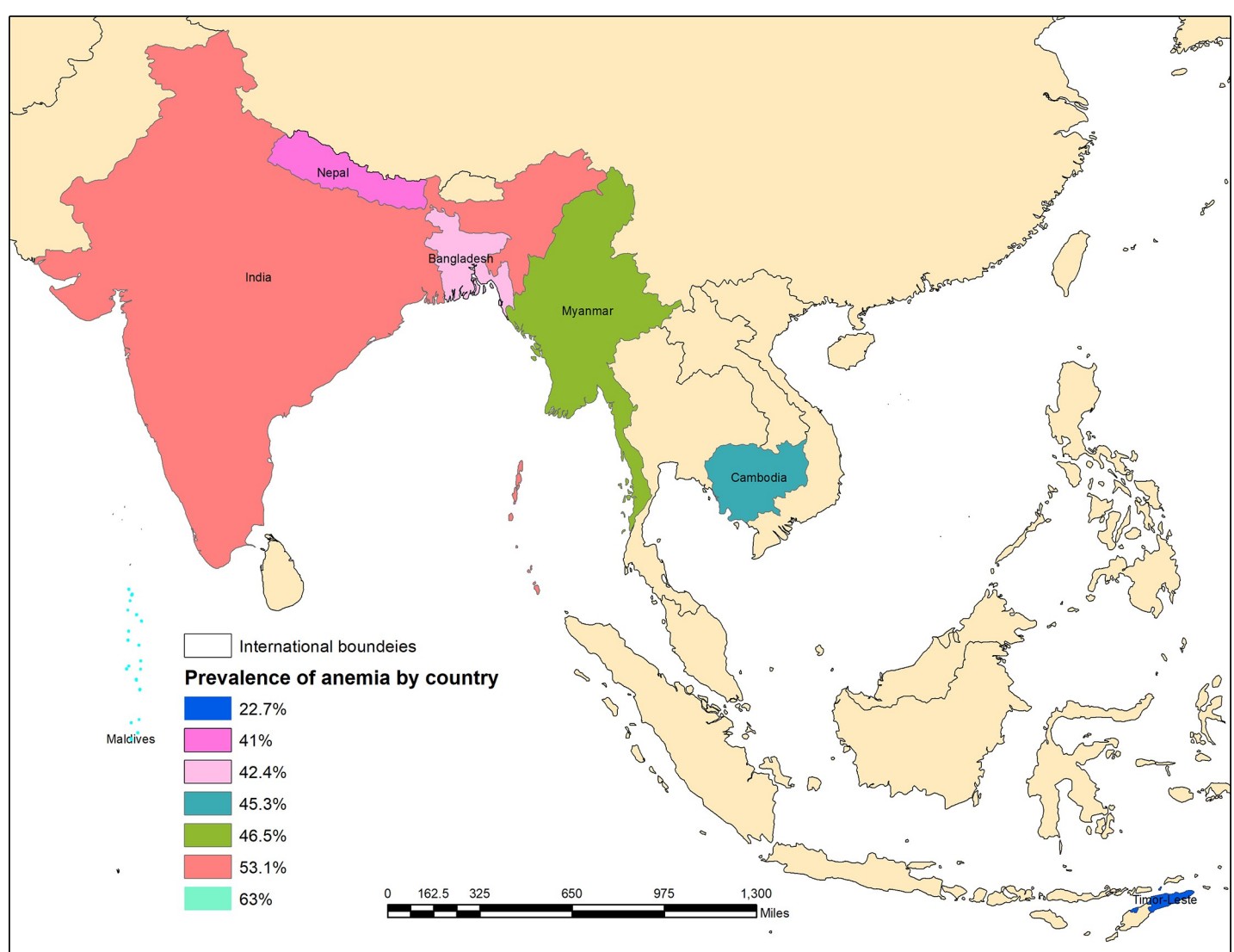

**Fig 1. Map showing the prevalence of anemia among women of reproductive age in seven selected South and Southeast Asian countries.**

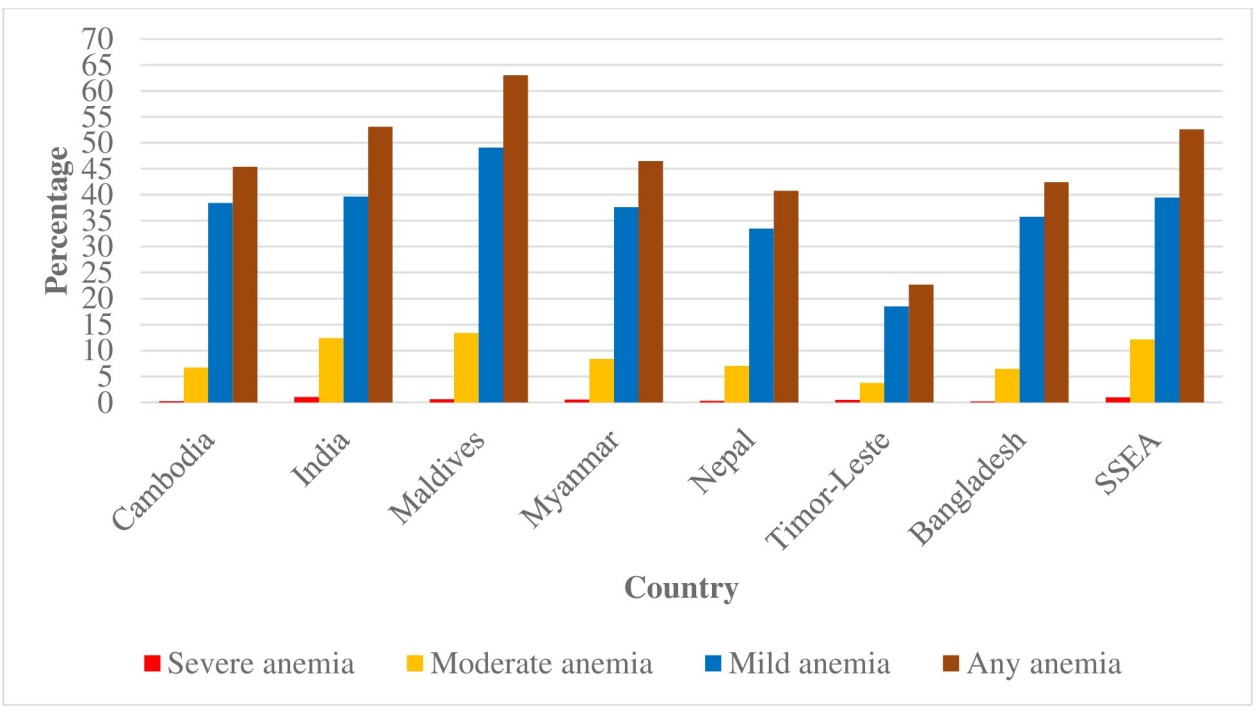

**Fig 2. Prevalence of any form of anemia among women of reproductive age by country.**

24.7% Vs 21.7%). In the pooled analysis, the prevalence of anemia was generally higher among the least educated women and the lower among the most educated women in Cambodia, India, Bangladesh, and Timor-Leste. However, in Maldives and Myanmar, the prevalence was highest in the most educated women. The prevalence of anemia among WRA was higher among the poorest economic class group in Cambodia (Poorest Vs richest: 51.1% Vs 40.9%), India (Poorest Vs richest: 56.8% Vs 49.6%), Bangladesh (Poorest Vs richest: 48.9% Vs 36.4%) and Mynamar (Poorest Vs richest: 47.5% Vs 45.2%). While the prevalence of anemia was highest in the middle-class group in Nepal and Timor-Leste. The prevalence of anemia was highest among the widow/divorced/separated women in all countries except Nepal and Myanmar where prevalence was higher among married women. Likewise, the prevalence was higher among those women who did not use not improving toilet facilities in all countries. Among WRA, the prevalence of anemia was highest among older age groups (≥35 years) in Cambodia (47.1%), Maldives (64.3%), Bangladesh (42.9%), and Myanmar (48.1%). The prevalence of anemia was highest among underweight nutritional status women in all countries. Among WRA, the prevalence of anemia was higher in those women who didn't consume iron supplementation during most recent pregnancy in all countries. Similarly, the prevalence was highest among those women who didn't ANC visit once in all countries except in Myanmar and Timor-Leste. The prevalence of anemia was highest among the women's having more than four children in Cambodia, India, Bangladesh and Maldives, and Nepal.

## Factors associated with anemia among women of reproductive age

Various factors were associated with anemia in each country (Table 2). In environmental and household factors, women's having no education had higher odds of anemia compared to those who have attended higher level of education in India (aOR = 1.24), but in Cambodia (aOR = 1.37) and Timor-Leste (aOR = 1.47), women's who attended at least primary level of

**Table 2. Multiple logistic regression analysis for factors associated with anemia among women of reproductive age in seven South and Southeast Asian countries.**

| Study variables | Cambodia | India | Maldives | Myanmar | Nepal | Timor-Leste | Bangladesh |
|---|---|---|---|---|---|---|---|
| | aOR(95% CI) | aOR(95% CI) | aOR(95% CI) | aOR(95% CI) | aOR(95% CI) | aOR(95% CI) | aOR(95% CI) |
| **Environmental and household factors[a]** | | | | | | | |
| **Residence** | | | | | | | |
| Urban | Ref | Ref | Ref | Ref | Ref | Ref | Ref |
| Rural | 1.12(0.98–1.27) | 1.02(0.98–1.05) | 0.33(0.22–0.48)*** | 1.01(0.84–1.20) | 1.07(0.89–1.28) | 0.77(0.63–0.95)* | 1.20(0.95–1.51) |
| **Education** | | | | | | | |
| No education | 1.29(0.97–1.69) | 1.24(1.18–1.30)*** | 0.72(0.32–1.57) | 0.91(0.71–1.07) | 1.01(1.80–1.27) | 1.26(0.87–1.82) | 1.36(0.94–1.95) |
| Primary | 1.37(1.07–1.75)* | 1.11(1.06–1.18)*** | 0.93(0.61–1.42) | 0.95(0.77–1.17) | 0.93(0.72–1.20) | 1.47(1.01–2.14)* | 1.41(0.98–2.02) |
| Secondary | 1.19(0.94,1.49) | 1.11(1.06–1.15)*** | 0.75(0.53–1.28) | 0.86(0.72–1.03) | 1.15(0.93–1.41) | 1.20(0.87–1.64) | 1.13(0.82–1.56) |
| Higher | Ref | Ref | Ref | Ref | Ref | Ref | Ref |
| **Wealth status** | | | | | | | |
| Poor | 1.38(1.17–1.64)*** | 1.15(1.10–1.19)*** | 1.61(1.05–2.47)* | 1.18(1.01–1.39)* | 0.77(0.65–0.91)** | 0.89(0.69–1.16) | 1.35(1.07–1.70)** |
| Middle | 1.08(0.92–1.27) | 1.02(0.98–1.05) | 1.54(1.01–2.35)* | 1.15(0.96–1.36) | 1.10(0.91–1.34) | 0.97(0.76–1.26) | 1.19(0.92–1.55) |
| Rich | Ref | Ref | Ref | Ref | Ref | Ref | Ref |
| **Marital status** | | | | | | | NA |
| Never in union | Ref | Ref | Ref | Ref | Ref | Ref | |
| Married/living together | 0.84(0.75–0.94)** | 0.97(0.94–1.00) | 1.33(1.00–1.77)* | 0.78(0.70–0.87)*** | 0.87(0.75–1.01) | 1.22(1.01–1.47)* | |
| Widowed/divorced/separated | 1.04(0.85–1.25) | 0.99(0.92–1.06) | 1.99(1.13–3.48) | 0.92(0.75–1.12) | 0.62(0.41–0.94)* | 1.73(1.04–2.89)* | |
| **Occupation** | | | | | | | |
| Not working | Ref | Ref | Ref | Ref | Ref | Ref | NA |
| Working | 0.93(0.82–1.05) | 1(0.98–1.03) | 0.97(0.80–1.17) | 0.97(0.86–1.09) | 0.96(0.82–1.11) | 0.97(0.81–1.15) | |
| **Toilet type[¥]** | | | | | | | |
| Improved | Ref | Ref | Ref | Ref | Ref | Ref | Ref |
| Not improved | 1(0.89–1.12) | 1.14(1.10–1.18)*** | 2.09(0.64–6.78) | 1.03(0.91–1.15) | 1.63(1.36–1.96)*** | 1.34(1.11–1.63)** | 0.99(0.77–1.26) |
| **Water source[€]** | | | | | | | |
| Improved | | Ref | Ref | Ref | Ref | Ref | Ref |
| Not improved | | 1.12(1.09–1.16)*** | 0.62(0.39–0.97)* | 0.95(0.86–1.06) | 1.59(1.34–1.88) | 1.11(0.93–1.33) | 1.43(0.96–2.13) |
| **Individual factors[b]** | | | | | | | |
| **Age group (years)** | | | | | | | |
| 15–24 | 1.04(0.85–1.25) | 1.07(1.05–1.09)*** | 0.78(0.61–0.99)* | 0.85(0.69–1.06) | 0.92(0.72–1.17) | 1.74(1.29–2.34)*** | 1.16(0.94–1.42) |
| 25–34 | Ref | Ref | Ref | Ref | Ref | Ref | Ref |
| ≥35 | 1.26(1.00–1.59)* | 1.03(0.99–1.07) | 0.99(0.79–1.23) | 1.05(0.85–1.28) | 0.63(0.43–0.94)* | 1.16(0.83–1.62) | 0.95(0.62–1.46) |
| **BMI** | | | | | | | |
| Normal | Ref | Ref | Ref | Ref | Ref | Ref | Ref |
| Underweight | 1.36(1.05–1.77)* | 1.31(1.27–1.33)*** | 2.43(1.55–3.81)*** | 1.16(0.89–1.51) | 1.18(0.91–1.53) | 1.05(0.79–1.40) | 1.24(0.97–1.57) |
| Overweight | 0.74(0.58–0.94)* | 0.76(0.74–0.78)*** | 0.80(0.66–0.97)* | 0.65(0.52–0.80)*** | 0.60(0.45–0.81)** | 0.84(0.56–1.26) | 0.61(0.42–0.90)* |
| Obesity | 0.74(0.39–1.40) | 0.76(0.73–0.81)*** | 0.82(0.65–1.03) | 0.39(0.27–0.58)*** | 0.85(0.44–1.63) | 1.31(0.59–2.88) | 0.76(0.38–1.52) |

*(Continued)*

**Table 2.** (Continued)

| Study variables | Cambodia | India | Maldives | Myanmar | Nepal | Timor-Leste | Bangladesh |
|---|---|---|---|---|---|---|---|
| | aOR(95% CI) | aOR(95% CI) | aOR(95% CI) | aOR(95% CI) | aOR(95% CI) | aOR(95% CI) | aOR(95% CI) |
| **Iron intake** | | | | | | | |
| No | 0.85(0.56–1.28) | 1.01(0.99–1.04) | - | 1.13(0.94–1.51) | 1.28(0.79–2.05) | 1.11(0.86–1.41) | 1.69(0.90–3.15) |
| Yes | Ref | Ref | | Ref | Ref | Ref | Ref |
| **Births in the last 5 years** | | | | | | | |
| No | Ref | Ref | Ref | Ref | Ref | Ref | Ref |
| 1 | - | - | - | - | - | - | - |
| >1 | 1.19(0.93–1.50) | 1.16(1.13–1.18)*** | 1.17(0.92–1.47) | 1.15(0.92–1.44) | 1.40(1.08–1.80)** | 1.22(0.95–1.56) | 1.29(1.00–1.66)* |
| **ANC visits during pregnancy** | | | | | | | |
| No | 1.03(0.69–1.52) | 1.08(1.04–1.11)*** | 2.10(0.93–4.73) | 0.81(0.59–1.09) | 1.06(0.67–1.67) | 1.04(0.73–1.48) | 1.12(0.79–1.58) |
| <5 times | 1.10(0.95–1.31) | 1.07(1.03–1.08)*** | 1.23(0.85–1.82) | 0.82(0.89–1.22) | 0.94(0.71–1.22) | 1.09(0.84–1.41) | 1.05(0.76–1.46) |
| ≥5 times | Ref | ref | Ref | Ref | Ref | Ref | Ref |
| **Currently breastfeeding** | | | | | | | |
| No | 0.70(0.59–0.83)*** | 0.92(0.90–0.94)*** | 1.23(1.03–1.45)* | 0.74(0.63–0.87)*** | 0.91(0.69–1.18) | 1.02(0.79–1.29) | 0.65(0.53–0.80)*** |
| Yes | Ref | Ref | Ref | Ref | Ref | Ref | Ref |
| **Total children ever born** | | | | | | | |
| No | Ref | Ref | Ref | Ref | Ref | Ref | Ref |
| 1–4 | - | - | - | - | - | - | - |
| >4 | 1.10(0.77–1.57) | 1.09(1.05–1.14)*** | 0.85(0.57–1.26) | 1.51(1.16–1.98**) | 1.33(0.86–2.04) | 1.19(0.86–1.65) | 1.31(0.91–1.87) |

aOR: adjusted odds ratio for the logistic regression.

[a]denotes for the adjusted prevalence ratios and 95% confidence intervals for the logistic regression model which included environmental and household factors.

[b]denotes for the adjusted prevalence ratios and 95% confidence intervals for the logistic regression model which included individual factors.

***p<0.001

**p<0.01

*p<0.05.

¥Improved toilet facility (flush toilet, piped sewer system, septic tank, flush/pour flush to pit latrine, ventilated improved pit latrine, pit latrine with slab, and composting toilet) [36].

€Improved water source (piped water into dwelling, piped water to yard/plot, public tap or standpipe, tube well or borehole, protected dug well-protected spring and rainwater) [36].

education were more likely to be anemic. Women who had poorer wealth status were more likely to be anemic compared to women with richer wealth status in Cambodia (aOR = 1.38), India (aOR = 1.15), Maldives (aOR = 1.61), Myanmar (aOR = 1.18) and Bangladesh (aOR = 1.35), but were less likely to be anemic in Nepal (aOR = 0.77). Women having widowed/divorced/separated were more likely to be anemic compared to never union women in Timor-Leste (aOR = 1.73), but higher odds of the increased risk of anemic among Maldivian women (aOR = 1.33). Conversely, married women were less likelihood of anemia in Cambodia (aOR = 0.84) and Myanmar (aOR = 0.78). Women who had a household with not improved toilet facilities were higher odds of anemia compared to improved toilet facilities in India (aOR = 1.14), Nepal (aOR = 1.63), and Timor-Leste (aOR = 1.34). Likewise, women who did not improve water sources were more likely to be anemic compared to those with an improved water source in India (aOR = 1.12).

In terms of individual factors among these populations, numerous factors were more likely to be anemic in most countries. The younger women (15–24 years) were more likely to be anemic compared to women in the age group of 25–34 years in India (aOR = 1.07) and Timor-Leste (aOR = 1.74), but ≥35 years age groups were less likely to be anemic in Nepal (aOR = 0.63), and Cambodia (aOR = 1.26). Being underweight (BMI<18.5 kg/m2) among WRA was more likely to be an increased risk of anemia compared to normal BMI in Cambodia (aOR = 1.36), India (aOR = 1.31) and Maldives (aOR = 2.43). Overweight and obesity were protective against the increased likelihood of anemia in Cambodia (aOR = 0.74), India (aOR = 0.76), Maldives (aOR = 0.80) Myanmar (aOR = 0.65), Nepal (aOR = 0.60) and Bangladesh (aOR = 0.61). Women who had more than one year of birth spacing were more likely to be anemic compared to no birth spacing in India (aOR = 1.16), Nepal (aOR = 1.40), and Bangladesh (aOR = 1.29). Among WRA, who never ANC visits during their pregnancy were more likely to be anemic compared to more than five times ANC visits in India (aOR = 1.08). Women who didn't currently breastfeed were more likely to be anemic than those with currently breastfeeding in Myanmar (aOR = 1.23), but not in Cambodia, India, Bangladesh, and the Maldives. Women who have more than four children were more likely of anemic compared to those with no children in India (aOR = 1.09) and Myanmar (aOR = 1.51).

## Discussion

This study provides country-level prevalence and associated factors of anemia among WRA in seven selected SSEA countries. The prevalence of anemia was 52.6%, ranged from 22.7% in Timor-Leste to 63% in the Maldives. According to WHO guidelines, the prevalence of anemia with the level above 40% among WRA is a serious public health problem [37]. This reflect anemia as a major public health problem in SSEA countries [38]. In our study, the prevalence of anemia varied from country to country. The possible reason may be characterized by the different dietary patterns, geographical, and cultural factors in these countries [13]. The systematic analysis reported the prevalence of anemia among non-pregnant women in South Asian countries declined slightly from an estimated 53% in 1995 to an estimated 47% in 2011, whilst the prevalence of anemia among pregnant women in the same region was almost stagnant (53% in 1995 and 52% in 2011) [2]. In Southeast Asian countries, the prevalence of anemia was 21% in non-pregnant and 25% in pregnant women [2]. We assume that these findings are consistent with our results because four out of seven countries in this study were from South Asia with a larger representation and had a higher burden of anemia among WRA [2,3]. High prevalence of anemia among women from this region are attributed by the social and biological vulnerability within both household and community [39]. In SSEA countries, along with the nutritional anemia poverty and gender inequality also play a significant role in contributing anemia [3].

The present study found the various household factors: women with low socioeconomic status, lack of education, not improved toilet facilities and water sources were an important predictor of anemia in the SSEA countries, which is similar to determinants of anemia in studies from Bangladesh [40] Timor-Leste [41] India [42]. These results are also consistent with a multi-country study across the LMICs which found that women's years of education, wealth status, cultural norms, values, and behavior were important predictors of anemia among women [3]. The systematic review conducted across the low-income countries exhibited that the education level along with wealth and cultural norms and behavior were overarching determinants of anemia among women [6]. Yang et al [39] also reported that the prevalence of anemia was observed higher among those individuals with low socioeconomic status from LMICs. In contrast, women in poorer wealth status were less likely to be anemic in Nepal. The

possible reason could be Nepal is an agrarian-based country and the eating pattern is almost the same for all [43]. Most of the Nepalese people consume iron-rich staple foods regardless of wealth status [43].

Women's level of education was not a determinant of anemia in Nepal, Maldives, and Myanmar. These findings are similar to Harding et al [11] results who found that women's education was not likely to be anemic in Nepal and Pakistan. Among women who belong to rural areas were less likely to be suffering from anemia in Maldives and Timor-Leste. Anemia is more prevalent in city areas than those with the rural areas in both countries (In Maldives: 73% in urban and 56% in rural areas and In Timor-Leste: 25% in urban and 22% in rural areas) [27,28]. According to the DHSs reports, Women living in urban areas with wealthier wealth status tended to increase anemia in these two countries [27,28]. The convergent findings could be attributed to the selection of apparently healthy women living in rural areas and non-endemic areas for malaria and hookworm infestation prone areas in both countries. Moreover, in the Maldives the traditional staple foods consumed in rural households are more diversified and rich in iron contains compared to other countries in the same region. Such diversified food patterns in the rural areas of Maldives might have a positive role in meeting the adequacy of iron requirements for the reproductive age of women [44,45].

This study found the prevalence of anemia was more prevalent in those households with not having improved toilets and water facilities. These findings are in line with the results from the nationally representative DHS dataset, where lack of toilet facility and water facility was more tended to increase the risk of anemia in Nepal, Pakistan, and Bangladesh [11,46]. In this study, women with underweight nutritional status ($<18.5$ kg/m$^2$ BMI) were more likely to be anemic in most of the South Asian countries. These findings are in line with a study from India [47], Bangladesh [40], Nepal, and Pakistan [11]. The possible reason could be exhibited by the fact that malnourished women have a greater chance of iron deficiency anemia which is usually associated with poor wealth status [48]. Our study found that the mothers who breastfeed their children are less likely to suffer from anemia in the case of Cambodia, India, Myanmar, and Bangladesh except for Maldives, Nepal, and Timor-Leste. Exclusive breastfeeding may reduce the anemia as it reduces the return of the menstrual cycle to 20–30 weeks. The lactating mothers who breastfed for two years had the lower odds of anemia than the mother who breastfed for one year's [49].

This study comprises of some limitations. First, due to the cross-sectional nature of data, it could not establish the causal pathway of the association between the predictors and explanatory variables. Second, comparable data on anemia among WRA were not accessible from all South and Southeast Asian countries, thus our analysis was limited to the selected seven South and Southeast Asian countries. Third, this study could not consider the assessment of dietary factors among women of reproductive age that might have attributed to the risk of development of anemia. Fourth, the DHSs data used from the seven different countries were taken from different years and periods between the surveys. Despite these limitations, the strengths of this study were the use of a multi-country population-based nationally representative samples. This study also provides important information on the prevalence and factors associated with anemia among WRA across this region using comparable datasets.

## Conclusions

This study highlighted a high prevalence of anemia among WRA in seven selected SSEA countries. Multiple factors related to household, environmental, and individual were associated with the continued rises of anemia among WRA. Improved water and sanitation status, completion of recommended ANC visits, normal BMI status, and better household income status

are crucial in reducing anemia in the SSEA region. The findings of the study urge the realistic multi-pronged approach to tackle the high prevalence of anemia across the countries in the region. Likewise, among WRA from low socio-economic status, having low nutrition education should be given high priority during both nutrition-sensitive and specific program implementation. Also, the existing national policies and programs need to be reviewed based on recent evidence to track the progress in meeting the WHO global nutrition targets 2025 and nutrition targets of SDGs 2030.

## Supporting information

**S1 Table. Data sources and sample size.**
(DOCX)

## Acknowledgments

We would like to thank the DHS program, ICF international for providing us the data set for analysis.

## Author Contributions

**Conceptualization:** Dev Ram Sunuwar, Devendra Raj Singh, Pranil Man Singh Pradhan.

**Data curation:** Dev Ram Sunuwar, Pushpa Rai.

**Formal analysis:** Dev Ram Sunuwar.

**Methodology:** Dev Ram Sunuwar, Narendra Kumar Chaudhary.

**Project administration:** Dev Ram Sunuwar, Devendra Raj Singh, Narendra Kumar Chaudhary, Pushpa Rai.

**Software:** Dev Ram Sunuwar.

**Supervision:** Dev Ram Sunuwar, Devendra Raj Singh, Narendra Kumar Chaudhary, Pranil Man Singh Pradhan, Pushpa Rai, Kalpana Tiwari.

**Validation:** Dev Ram Sunuwar, Devendra Raj Singh.

**Visualization:** Dev Ram Sunuwar, Devendra Raj Singh.

**Writing – original draft:** Dev Ram Sunuwar, Devendra Raj Singh, Narendra Kumar Chaudhary, Pranil Man Singh Pradhan, Pushpa Rai, Kalpana Tiwari.

**Writing – review & editing:** Dev Ram Sunuwar, Devendra Raj Singh, Narendra Kumar Chaudhary, Pranil Man Singh Pradhan, Pushpa Rai, Kalpana Tiwari.

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
