## [Decision Letter · Decision Letter 0]

29 May 2020

PONE-D-20-10365

Prevalence and factors associated with anemia among women in seven South and Southeast Asian Countries: evidence from nationally representatives survey

PLOS ONE

Dear Dr. Sunuwar,

Thank you for submitting your manuscript to PLOS ONE. After careful consideration, we feel that it has merit but does not fully meet PLOS ONE’s publication criteria as it currently stands. Therefore, we invite you to submit a revised version of the manuscript that addresses the points raised during the review process.

The reviewers have pointed out many important aspects for revision of the manuscript. A major improvement is necessary for further evaluation of the revised manuscript. 

We look forward to receiving your revised manuscript.

Kind regards,

Marly A. Cardoso, Ph.D.

Academic Editor

PLOS ONE

Journal Requirements:

3. We note that Figure 1 in your submission contain map images which may be copyrighted. All PLOS content is published under the Creative Commons Attribution License (CC BY 4.0), which means that the manuscript, images, and Supporting Information files will be freely available online, and any third party is permitted to access, download, copy, distribute, and use these materials in any way, even commercially, with proper attribution. For these reasons, we cannot publish previously copyrighted maps or satellite images created using proprietary data, such as Google software (Google Maps, Street View, and Earth). For more information, see our copyright guidelines: http://journals.plos.org/plosone/s/licenses-and-copyright.

a) You may seek permission from the original copyright holder of Figure(s) [#] to publish the content specifically under the CC BY 4.0 license. 

Reviewers' comments:

Reviewer's Responses to Questions

**Comments to the Author**

1. Is the manuscript technically sound, and do the data support the conclusions?

Reviewer #1: Yes

Reviewer #2: Yes

2. Has the statistical analysis been performed appropriately and rigorously? 

Reviewer #1: Yes

Reviewer #2: Yes

3. Have the authors made all data underlying the findings in their manuscript fully available?

Reviewer #1: Yes

Reviewer #2: Yes

4. Is the manuscript presented in an intelligible fashion and written in standard English?

Reviewer #1: Yes

Reviewer #2: No

5. Review Comments to the Author

Reviewer #1: This study aimed to assess the prevalence and factors associated with anemia among women of reproductive age (WRA) in seven selected South and Southeast Asian countries.

This study was a secondary analysis of the seven selected South and Southeast Asian countries. During the 2012 World Health Assembly, the World Health Organization endorsed a target of a 50% reduction of anemia among women of reproductive age by 2025. Therefore, the study deals with a well-updated subject.

Very few studies have utilized nationally representative data to investigate the prevalence and determinants of anemia among reproductive-aged women in the South and Southeast Asian context. With this, it adds important data.

Here are suggestions for the article.

Title

Suggestion: "Prevalence and factors associated with anemia among women of reproductive age in seven South and Southeast Asian Countries: evidence from nationally representatives survey".

The insertion of the term "women of reproductive age" better specifies the study population. And, it would not lead to exceed the number of characters allowed.

Abstract

In lines 24, 28 and 31 of the Abstract section there is an abbreviation, which is not prohibited, but should be avoided in this part of the text, according to the rules of the journal. The inclusion of the term "women of reproductive age" in its entirety would not exceed the number of characters in the section (300). Or, this can be a term added in the title, also without exceeding the character limit of the title which would better define the population studied.

The keywords "associated factors" and "South Southeast Asia" were not found in the MESH descriptors.

The text is brief, presents the objective of the study, includes substantial results, focuses on positive and non-negative findings.

Introduction

The authors start from the macro idea, contextualize, and even identify the problem. The section is finished with the justification and the main objective of the work.

Methodology

Experiments, statistics, and other analyses were performed to a high technical standard and were described in sufficient detail.

Line 141 – change the unit gm/dl by g/dL for both hemoglobin <12g/dL and 11g/dL.

Line 142 – change Homocue for Hemocue.

In lines 146, 147 and 148, the authors say that the predictor variables were chosen according to the literature review. However, in the introduction they say that several approaches at the population level have been taken over the years, such as micronutrient supplementation among adolescent girls and women, food fortification, nutritional education, counseling and orientation of an iron-rich dietary plan for at-risk populations. However, these are not variables evaluated in the study.

In lines 154 and 155 the authors need to explain better that the "wealth index" is a measure calculated by DHS itself and that it is based on the possession of consumer goods. It was not clear how it was obtained and, above all, how the categories were categorized. The understanding that it is a variable calculated by DHS is only possible if the reader searches the DHS Program website further.

Line 159 defines abbreviations upon first appearance in the text of ANC.

The variable iron intake is mentioned in lines 158 and 162, but do not explain what type of iron intake was this: supplementation acquired by women, given by some public policy, homemade fortification, dietetics? In the discussion (line 337), the limitation of not having evaluated food consumption data is mentioned. This variable is external intake? It is not clear.

The research meets all applicable standards for the ethics of experimentation and research integrity.

Results

Table 2 -age group years - specify where women aged 35 entered. If it is in the third category express as ≥35 years.

Discussion

In line 297, a study that demonstrates the prevalence of anemia in an age group quite different from that of the study (children).

In lines 314 to 319, the authors return to the results of anemia prevalence in rural and urban areas, present a convergent reference to the finding, but do not discuss the possible causes of this.

In line 337, when they mention the absence of food intake data, they speak only of the intake of "mothers and children". They do not talk about the intake of non-pregnant women and the children's food intake would not reflect the mothers' food intake, because the eating pattern of such opposite age groups is different.

Conclusion

The conclusion section responds to the proposed objectives and closes the subject by resuming the justification of the study and makes a brief prospective observation of what should be done from the findings.

References

Almost all references are from the last 10 years.

Review formatting of article references by placing them in Vancouver style. Examples of changes that need to be made include:

- The issues of journals should come in parentheses - examples - References 12, 15, 18, 19, 20.

- You do not put pp. before referring to the pages of journals.

- Reference 20 - quote in the supplement standards.

- References 14 and 25 do not contain the names of journals.

- On the pages of the references use only the unit or dozen on the final pages mentioned.

The article can be published after the mentioned corrections.

Reviewer #2: PARECER PONE-D-20-10365

Prevalence and factors associated with anemia among women in seven South and Southeast Asian Countries: evidence from nationally representatives survey

The study aimed to assess the prevalence and factors associated with anemia in women of reproductive age in seven selected countries in South and Southeast Asia. Based on the most recent data obtained from the Demographic and Health Surveys (DHS) of these seven countries, they analyzed data from 726,164 women. Multivariable binary logistic regression was performed to assess the factors associated with anemia among women in each country separately. Therefore, it is of relevance to public health study, which used nationally representative sample and appropriate statistical analysis. With minor adjustments it will be in conditions of publication.

The introduction presents prolix writing and with redundant information or that should be in another section. For example, right in the second line of the introduction it defines the cutoff points for anemia, which is suitable for is in methods (as it is also). There is an exaggerated exploration of the factors associated with anemia, as well as prevalence values in different situations. Much of this information would be better used in the discussion, situating the results found.

Line 159 – “…ever terminated pregnancy, ANC during pregnancy”: ANC must be defined a priori.

Line 160-161 – delete the second BMI: BMI was categorized into underweight BMI...

Line 167 – “…analyzed using STATA/MP version 14.1” / Stata/MP, version 14.1

Line 190 – It refers to figure 1, but the figure shown is named as figure 2. By the way, the title of this figure should be more intelligible (self-sufficient), so that the reader can understand what it is about without having to resort to the text.

Line 219 – “...women who didn't consume iron in all countries.” Use more precise wording: although insufficient, some amount they consumed.

Line 227 – In the table title, write WRA in full.

Lines 247-248 – “The association of anemia among 15-49 years old women with environmental and individual factors was examined using the binary logistic regression model.” This has already been reported in methods.

Line 249 - Women's having... / women's having...

Lines 265-266 – “age groups 15-24 years was positively associated with an increased likelihood of anemia compared to age groups of 25-34…” the term positively seems misused. If the variables are categorical (and not ordinal or continuous) and the highest prevalence was in the youngest age group, how was the relationship positive? He's confused. Confirm the adequacy of this term in other parts of the article.

Line 296-301 – “The cross-sectional study from the DHS done in 27 Sub-Saharan Africa (SSA) between 2008 and 2014 found that the prevalence of anemia among children was 59.9% [33], which is slightly higher than our findings.” It makes no sense to compare data from children with those from women. If you want to use the information, make some bridge to make sense.

Line 333 – “This study comprises of some limitations. First, due to the cross-sectional nature of data, this study…” Delete the last this study.

Line 335 – “Second, Comparable data. / Second, comparable data…

Line 337 – “Third, dietary intake of mothers and children were not assessed…” the study does not address children.

6. PLOS authors have the option to publish the peer review history of their article (what does this mean?). If published, this will include your full peer review and any attached files.

Reviewer #1: No

Reviewer #2: Yes: HAROLDO DA SILVA FERREIRA

---

## [Author Response · Author response to Decision Letter 0]

11 Jun 2020

Manuscript ID: PONE-D-20-10365

Manuscript title: Prevalence and factors associated with anemia among women of reproductive age in seven South and Southeast Asian Countries: evidence from nationally representative surveys

Dear Editor, 

We would like to thank the reviewers for the specific and needful comments. We have modified the paper in response to the extensive and insightful reviewers' comments. Sincerest thanks for providing us an opportunity to submit a revised copy of our manuscript entitled "Prevalence and factors associated with anemia among women of reproductive age in seven South and Southeast Asian Countries: evidence from nationally representative surveys" to esteemed Journal PLOS ONE. We appreciate the time and efforts that you and the reviewers have dedicated to providing your valuable feedback on our manuscript. We are grateful to the reviewers for their insightful comments on our papers. We believe the revisions have substantially improved our manuscript. 

 We have highlighted the changes within the manuscript where necessary. We hope that the manuscript is now suitable for publication. We look forward to hearing a positive response from you.

Here is a point-by-point response to the academic editor and reviewers' comments and concerns. 

Academic editor comments and concern

Response: Thank you for your suggestions. We have revised the formatting of our manuscript as per the PLOS ONE's manuscript body formatting guidelines. 

Response: Thank you for your valuable suggestions and comments. We used data of Demographic and Health Surveys that are publicly available and can be freely downloaded upon the formal request from the DHS website (https://www.dhsprogram.com/data/available-datasets.cfm). As per the guidelines and regulations of DHS, the dataset obtained by the authors for the purpose of statistical reporting and analysis specifically for this paper is not allowed to share in the public domain. However, we have uploaded the modified datasets in the online manuscript submission system to be only shared with the journal editor/reviewers for the purpose of the manuscript review process. 

 3. We note that Figure 1 in your submission contain map images which may be copyrighted. All PLOS content is published under the Creative Commons Attribution License (CC BY 4.0), which means that the manuscript, images, and Supporting Information files will be freely available online, and any third party is permitted to access, download, copy, distribute, and use these materials in any way, even commercially, with proper attribution. For these reasons, we cannot publish previously copyrighted maps or satellite images created using proprietary data, such as Google software (Google Maps, Street View, and Earth). For more information, see our copyright guidelines: http://journals.plos.org/plosone/s/licenses-and-copyright.

Response: Thank you very much for pointing out the important aspects of our manuscript. We assure you that the provided Fig1 contains a study area Map with anemia prevalence is copyright free. We created this map using Arc GIS software version 10.8 and base file of the administrative national and subnational bounderies for seven selected South and Southeast Asian (SSEA) country were obtained from the freely available copyright free resources Natural Earth (http://www.naturalearthdata.com/). The map was displayed by joining the prevalence of anemia with corresponding country. Also, we have mentioned this point within the manuscript, methods section of page number 9, line 187-189 within the revised manuscript.

Reviewers' comments:

Reviewer #1

Comment: This study aimed to assess the prevalence and factors associated with anemia among women of reproductive age (WRA) in seven selected South and Southeast Asian countries. This study was a secondary analysis of the seven selected South and Southeast Asian countries. During the 2012 World Health Assembly, the World Health Organization endorsed a target of a 50% reduction of anemia among women of reproductive age by 2025. Therefore, the study deals with a well-updated subject. Very few studies have utilized nationally representative data to investigate the prevalence and determinants of anemia among reproductive-aged women in the South and Southeast Asian context. With this, it adds important data.

Here are suggestions for the article.

Response: Thank you very much for your time spent and efforts to provide valuable comments on our manuscript. 

Comments: Title Suggestion: "Prevalence and factors associated with anemia among women of reproductive age in seven South and Southeast Asian Countries: evidence from nationally representatives survey".

The insertion of the term "women of reproductive age" better specifies the study population. And, it would not lead to exceed the number of characters allowed.

Response: Thank you very much for your valuable suggestion. We have added the term "women of reproductive age" accordingly in the title and where necessary within the revised manuscript. 

Abstract

Comments: In lines 24, 28 and 31 of the Abstract section there is an abbreviation, which is not prohibited, but should be avoided in this part of the text, according to the rules of the journal. The inclusion of the term "women of reproductive age" in its entirety would not exceed the number of characters in the section (300). Or, this can be a term added in the title, also without exceeding the character limit of the title which would better define the population studied.

Response: Thank you for pointing this out. We have replaced the WRA with "women of reproductive age" in the abstract section and where necessary in our revised manuscript. 

Comments: The keywords "associated factors" and "South Southeast Asia" were not found in the MESH descriptors.

The text is brief, presents the objective of the study, includes substantial results, focuses on positive and non-negative findings.

Response: Thank you for your suggestion. According to the PLOS ONE's Journal manuscript body formatting guidelines, guidelines for keywords are not mentioned (https://journals.plos.org/plosone/s/file?id=wjVg/PLOSOne_formatting_sample_main_body.pdf). Also, most of the published articles have not used keywords in their article. Thus, in order to adhere to the manuscript body formatting guidelines, we removed all the keywords from the manuscript. However, we have submitted the appropriate keywords in the online manuscript submission form as per the PLOS one guideline. 

Introduction

comments: The authors start from the macro idea, contextualize, and even identify the problem. The section is finished with the justification and the main objective of the work.

Response: Thank you very much.

Methodology

Comments: Experiments, statistics, and other analyses were performed to a high technical standard and were described in sufficient detail.

Response: Thank you.

Comments: Line 141 – change the unit gm/dl by g/dL for both hemoglobin <12g/dL and 11g/dL. Line 142 – change Homocue for Hemocue.

Response: Thank you for your comments. We have replaced the unit gm/dl with "g/dL" where necessary within the manuscript. Also, we have replaced the word Homocue with "Hemocue" in line 144 within the revised manuscript. 

Comments: In lines 146, 147 and 148, the authors say that the predictor variables were chosen according to the literature review. However, in the introduction they say that several approaches at the population level have been taken over the years, such as micronutrient supplementation among adolescent girls and women, food fortification, nutritional education, counseling and orientation of an iron-rich dietary plan for at-risk populations. However, these are not variables evaluated in the study.

Response: Thank you. We agree with these constructive comments. We intended to explore the prevalence and factors associated with anemia among women of reproductive age in seven selected South and Southeast Asian countries. Based on the previous literature review regarding the risk of developing anemia among women in developing countries and the availability of data in the DHS dataset, we included household, environmental and individual factors as predictors' of anemia in our data analysis.. Meanwhile, we just underline the existing prevention and control strategy of anemia among women of reproductive age in the introduction section of our manuscript. These variables were not included because DHS does not collect this data. Even though these factors were not assessed in our study we believe highlighting these different approaches would provide the significance of the topic to the readers. 

Comments: In lines 154 and 155 the authors need to explain better that the "wealth index" is a measure calculated by DHS itself and that it is based on the possession of consumer goods. It was not clear how it was obtained and, above all, how the categories were categorized. The understanding that it is a variable calculated by DHS is only possible if the reader searches the DHS Program website further.

Response: Thank you for pointing this out. We have added the "wealth index calculation through principal component analysis (PCA) which is a standardized measure from household assets" [1]. "The wealth index variables were already included in the DHS dataset as five quintiles ranked as poorest, poorer, middle, richer, and richest. In this study, we further re-categorized as poor (poorest and poorer), middle and rich (richer and richest) for the analysis purpose" [2] in page 7 and 8, line 156-161 within the revised manuscript. 

Comments: Line 159 defines abbreviations upon first appearance in the text of ANC.

The variable iron intake is mentioned in lines 158 and 162, but do not explain what type of iron intake was this: supplementation acquired by women, given by some public policy, homemade fortification, dietetics? In the discussion (line 337), the limitation of not having evaluated food consumption data is mentioned. This variable is external intake? It is not clear.

We agree with these constructive comments. We have added the full form of ANC with "Antenatal checkup" in the manuscript in line 166 within the revised manuscript .

Yes, we have included iron supplementation during the most recent pregnancy in our analysis and revised the sentence accordingly in lines 170-171 within the revised manuscript. Indeed, Iron deficiency anemia is considered as one of the most common nutritional deficiency anemia among pregnant women. Iron folic acid supplementation program has been implemented in most of the south and Southeast Asian countries. Considering the significant role of iron-folic acid supplementation to prevent the anemia among women of reproductive age, we incorporated the iron intake variable in the analysis. 

Comments: The research meets all applicable standards for the ethics of experimentation and research integrity.

Response: Thank you very much. 

Results

comments: Table 2 -age group years - specify where women aged 35 entered. If it is in the third category express as ≥35 years.

Response: Thank you for pointing this out. We have corrected the age group years in Table 2 with ">35 years".

Discussion

comments: In line 297, a study that demonstrates the prevalence of anemia in an age group quite different from that of the study (children).

Response: We removed the word children.

Comments: In lines 314 to 319, the authors return to the results of anemia prevalence in rural and urban areas, present a convergent reference to the finding, but do not discuss the possible causes of this.

Response: We agree with your valuable comments. We have added the relevant discussion in our revised manuscript in line 335-340 as "These convergent findings could be due to the selection of apparently healthy women living in rural areas and non-endemic areas for malaria and hookworm infestation prone areas in both countries. Moreover, in the Maldives the traditional staple foods consumed in rural households are more diversified and rich in iron contains compared to other countries in the same region. Such diversified food patterns in the rural areas of Maldives might have a positive role in meeting the adequacy of iron requirements for the reproductive age of women"[3,4].

Comments: In line 337, when they mention the absence of food intake data, they speak only of the intake of "mothers and children". They do not talk about the intake of non-pregnant women and the children's food intake would not reflect the mothers' food intake, because the eating pattern of such opposite age groups is different.

Response: Thank you for your comments. We have revised the sentence with "this study could not consider the assessment of dietary factors among women of reproductive age that might have attributed to the risk of development of anemia" in line 358-359 within the revised manuscript.

Conclusion

comments: The conclusion section responds to the proposed objectives and closes the subject by resuming the justification of the study and makes a brief prospective observation of what should be done from the findings.

Response: Thank you very much.

References

comments: Almost all references are from the last 10 years.

Review formatting of article references by placing them in Vancouver style. Examples of changes that need to be made include:

- The issues of journals should come in parentheses - examples - References 12, 15, 18, 19, 20.

- You do not put pp. before referring to the pages of journals.

- Reference 20 - quote in the supplement standards.

- References 14 and 25 do not contain the names of journals.

- On the pages of the references use only the unit or dozen on the final pages mentioned.

Response: We thank the reviewer for these important suggestions. We have corrected the references accordingly in our revised manuscript where necessary. 

Reviewer #2:

Comments: Prevalence and factors associated with anemia among women in seven South and Southeast Asian Countries: evidence from nationally representatives survey

The study aimed to assess the prevalence and factors associated with anemia in women of reproductive age in seven selected countries in South and Southeast Asia. Based on the most recent data obtained from the Demographic and Health Surveys (DHS) of these seven countries, they analyzed data from 726,164 women. Multivariable binary logistic regression was performed to assess the factors associated with anemia among women in each country separately. Therefore, it is of relevance to public health study, which used nationally representative sample and appropriate statistical analysis. With minor adjustments it will be in conditions of publication.

Response: Thank you very much for your time spent and efforts to provide valuable comments on our manuscript. 

Comments: The introduction presents prolix writing and with redundant information or that should be in another section. For example, right in the second line of the introduction it defines the cutoff points for anemia, which is suitable for is in methods (as it is also). There is an exaggerated exploration of the factors associated with anemia, as well as prevalence values in different situations. Much of this information would be better used in the discussion, situating the results found.

Response: Thank you for your suggestion. We have revised the sentences accordingly in the introduction section of our revised manuscript. The cut-off points for anemia have been removed from the introduction section in line 48-49 and moved to the methods section accordingly in line 143-144 within the revised manuscript. 

Also, we added the information in discussion section as " Stevens et al., (2013) reported the prevalence of anemia among non-pregnant women in South Asian countries declined slightly from an estimated 53% in 1995 to an estimated 47% in 2011, whilst the prevalence of anemia among pregnant women in the same region was almost stagnant (53% in 1995 and 52% in 2011)" [5] in line 303-306 within the revised manuscript.

Comments: Line 159 – “…ever terminated pregnancy, ANC during pregnancy”: ANC must be defined a priori.

Response: We have revised and elaborate the sentence ever terminated pregnancy as "Whether the respondent ever had a pregnancy that terminated in a miscarriage, abortion, and stillbirth" in line 168-170 within the revised manuscript with track changes. Also, Antenatal checkup (ANC) during pregnancy refers to the "number of antenatal visits during pregnancy" in line 165-167 within the revised manuscript. 

Comments: Line 160-161 – delete the second BMI: BMI was categorized into underweight BMI...

Response: Thank you for pointing this out. We deleted the second BMI. 

Comments: Line 167 – “…analyzed using STATA/MP version 14.1” / Stata/MP, version 14.1

Response: We replaced the word STATA/MP with "Stata/MP version 14.1" accordingly in line 176.

Comments: Line 190 – It refers to figure 1, but the figure shown is named as figure 2. By the way, the title of this figure should be more intelligible (self-sufficient), so that the reader can understand what it is about without having to resort to the text.

Response: Thank you for your comments. We have revised the title Fig 1 with "Prevalence of anemia among women of reproductive age by the geographical locations in seven South and Southeast Asia countries” in line 200-201, which implied to display the geographical pattern and distribution of anemia prevalence rate using the map in the selected seven South and Southeast Asian country. Also, the map implies the study area being chosen for the analysis. Also, we have modified the Fig 2 with "Prevalence of any form of anemia among women of reproductive age by country" in line 235 within the revised manuscript.

Comments: Line 219 – “...women who didn't consume iron in all countries.” Use more precise wording: although insufficient, some amount they consumed.

Response: We have added the sentence in the line 230 within the revised manuscript with "iron supplementation during most recent pregnancy" to make more clear to the readers. 

Comments: Line 227 – In the table title, write WRA in full.

Response: Thank you for pointing out this. We have included the full form WRA with "women of reproductive age" in the revised title of both Table 1 and Table 2.

Comments: Lines 247-248 – “The association of anemia among 15-49 years old women with environmental and individual factors was examined using the binary logistic regression model.” This has already been reported in methods.

Response: Thank you for your suggestion. We have deleted this sentence accordingly. 

Comments: Line 249 - Women's having... / women's having...

Response: We have changed the word “Women's” with "women's" accordingly in the line 261. 

Comments: Lines 265-266 – “age groups 15-24 years was positively associated with an increased likelihood of anemia compared to age groups of 25-34…” the term positively seems misused. If the variables are categorical (and not ordinal or continuous) and the highest prevalence was in the youngest age group, how was the relationship positive? He's confused. Confirm the adequacy of this term in other parts of the article.

Response: We thank the reviewer for constructive comments. We have corrected the sentence in the line 277-279 within the revised manuscript with "the younger women (15-24 years) were more likely to be anemic compared with women in the age group of 25-34 years"[6].

Comments: Line 296-301 – “The cross-sectional study from the DHS done in 27 Sub-Saharan Africa (SSA) between 2008 and 2014 found that the prevalence of anemia among children was 59.9% [33], which is slightly higher than our findings.” It makes no sense to compare data from children with those from women. If you want to use the information, make some bridge to make sense.

Response: Thank you very much for your insightful comments. We have removed these sentences and added the more relevant evidence from the previous findings in the line 310-313 with “High prevalence of anemia among women from this region is attributed to the social and biological vulnerability within both household and community [7]. In SSEA countries, along with the nutritional anemia poverty and gender inequality also play a significant role in contributing anemia [8]” within the revised manuscript.

Comments: Line 333 – “This study comprises of some limitations. First, due to the cross-sectional nature of data, this study…” Delete the last this study.

Response: Thank you. We have deleted the “last this study" and revised the sentence to make it clearer in line 354-355 within the revised manuscript with track changes.

Comments: Line 335 – “Second, Comparable data. / Second, comparable data…

Response: We revised the word accordingly in line 356. 

Line 337 – “Third, dietary intake of mothers and children were not assessed…” the study does not address children.

Response: Thank you very for your constructive comments and feedback. We have revised the sentence with "this study could not consider the assessment of dietary factors among women of reproductive age that might have attributed to the risk of development of anemia" in line 358-359 within the revised manuscript. We removed the word "children" in line 363. 

We look forward to hearing a positive response from you soon.

Thank you. 

Sincerely

Mr. Dev Ram Sunuwar, Dietician/Researcher

Armed Police Force Hospital, Kathmandu, Nepal

Email: devramsunuwar@gmail.com

References

1. Rutstein SO, Johnson K. The DHS wealth index. Calverton, Maryland, USA: ORC Macro,. 2004. 

2. Gautam S, Min H, Kim H, Id HJ. Determining factors for the prevalence of anemia in women of reproductive age in Nepal : Evidence from recent national survey data. 2019; 1–17. 

3. Laxmi Pandey V, Mahendra Dev S, Jayachandran U. Impact of agricultural interventions on the nutritional status in South Asia: A review. Food Policy. 2016;62: 28–40. doi:10.1016/j.foodpol.2016.05.002

4. Nath P, P.B. Gaddagimath. Horticulture and Livelihood Security - Google Books. [cited 4 Jun 2020]. Available: https://books.google.com.np/books?id=pEJLDwAAQBAJ&pg=PA127&lpg=PA127&dq=iron+rich+food+in+rural+area+of+maldives&source=bl&ots=MTYgRIUWyO&sig=ACfU3U2ouD86JvsAP8ZNcgCb1NwAW8cJZQ&hl=en&sa=X&ved=2ahUKEwjz7oOmqeXpAhUCbisKHTXfAaUQ6AEwAXoECAoQAQ#v=onepage&q=iro

5. Stevens GA, Finucane MM, De-Regil LM, Paciorek CJ, Flaxman SR, Branca F, et al. Global, regional, and national trends in haemoglobin concentration and prevalence of total and severe anaemia in children and pregnant and non-pregnant women for 1995-2011: A systematic analysis of population-representative data. Lancet Glob Heal. 2013;1: e16. doi:10.1016/S2214-109X(13)70001-9

6. Kibret KT, Chojenta C, D’Arcy E, Loxton D. Spatial distribution and determinant factors of anaemia among women of reproductive age in Ethiopia: A multilevel and spatial analysis. BMJ Open. 2019;9: 1–14. doi:10.1136/bmjopen-2018-027276

7. Yang F, Liu X, Zha P. Trends in Socioeconomic Inequalities and Prevalence of Anemia Among Children and Nonpregnant Women in Low- and Middle-Income Countries. JAMA Netw open. 2018;1: e182899. doi:10.1001/jamanetworkopen.2018.2899

8. Kassebaum NJ, Jasrasaria R, Naghavi M, Wulf SK, Johns N, Lozano R, et al. A systematic analysis of global anemia burden from 1990 to 2010. Blood. 2014;123: 615–24. doi:10.1182/blood-2013-06-508325

---

## [Decision Letter · Decision Letter 1]

24 Jun 2020

PONE-D-20-10365R1

Prevalence and factors associated with anemia among women of reproductive age in seven South and Southeast Asian countries: evidence from nationally representative surveys

PLOS ONE

Dear Dr. Sunuwar,

Thank you for submitting your manuscript to PLOS ONE. After careful consideration, we feel that it has merit but does not fully meet PLOS ONE’s publication criteria as it currently stands. Therefore, we invite you to submit a revised version of the manuscript that addresses the points raised during the review process.

Please address all minor suggestions pointed out by the two reviewers. 

We look forward to receiving your revised manuscript.

Kind regards,

Marly A. Cardoso, Ph.D.

Academic Editor

PLOS ONE

Reviewers' comments:

Reviewer's Responses to Questions

**Comments to the Author**

1. If the authors have adequately addressed your comments raised in a previous round of review and you feel that this manuscript is now acceptable for publication, you may indicate that here to bypass the “Comments to the Author” section, enter your conflict of interest statement in the “Confidential to Editor” section, and submit your "Accept" recommendation.

Reviewer #1: All comments have been addressed

Reviewer #2: (No Response)

2. Is the manuscript technically sound, and do the data support the conclusions?

Reviewer #1: Yes

Reviewer #2: Yes

3. Has the statistical analysis been performed appropriately and rigorously? 

Reviewer #1: Yes

Reviewer #2: Yes

4. Have the authors made all data underlying the findings in their manuscript fully available?

Reviewer #1: Yes

Reviewer #2: Yes

5. Is the manuscript presented in an intelligible fashion and written in standard English?

Reviewer #1: Yes

Reviewer #2: No

6. Review Comments to the Author

Reviewer #1: I believe that almost all comments were answered properly, which made the article clearer. Therefore, it can be published.

I suggest that in reference number 14 the name of the journal BMC Women’s Health be included, as this suggestion has not been heeded. However, after this correction, there is no need to return the article to the reviewers for further evaluation.

Reviewer #2: PLOS ONE

Prevalence and factors associated with anemia among women of reproductive age in seven South and Southeast Asian countries: evidence from nationally representative surveys

PONE-D-20-10365R1

The authors responded to most of the suggestions presented in the previous version. However, there are still minor changes to be made before the manuscript is published. Next, I specify what these changes would be.

Line 33: The combined prevalence of anemia among women of reproductive age in the seven selected South and Southeast Asian countries was…

Change to: The combined prevalence of anemia was…

Line 28-30: (Bangladesh DHS 2011, Cambodia DHS 2014, India NFHS 2016, Maldives DHS 2016, Myanmar DHS 2015, Nepal DHS 2016, Timor-Leste 2015)...

Change to: (Bangladesh, Cambodia, India, Maldives, Myanmar, Nepal and Timor-Leste)...

Line 66: …conducted by Balarajan Y., et al (2011) reported that […] urban or rural settings [6].

Change to: …conducted by Balarajan et al. [6] reported that [...] urban or rural settings.

Line 113: Therefore, this study aims to identify […]: Start in a new paragraph.

Line 143: […] anemia was defined […]: Put a comma between g/dL and respectively: (g/dL, respectively)

Line 144: Hemoglobin level was assessed using capillary blood and the HemoCue rapid testing technique […]

Wouldn't that be better? Hemoglobin level in capillary blood was assessed using the HemoCue rapid testing technique […]

Line 173: […] than one year. ANC visits during […]. In Portuguese, does not start sentence with abbreviations or numbers. I am not sure if there is this rule in English (I am not an English speaker).

Line 206: Fig 1. Prevalence of anemia […]: Place after the next paragraph, in which fig. 1 is referred to for the first time.

Line 329: […]. Yang F et al. (2018) also reported […] middle-income countries [38].

Change to: […]. Yang et al. [38] also reported […] middle-income countries.

Other similar cases exist throughout the text and should be corrected.

7. PLOS authors have the option to publish the peer review history of their article (what does this mean?). If published, this will include your full peer review and any attached files.

Reviewer #1: No

Reviewer #2: Yes: HAROLDO DA SILVA FERREIRA

---

## [Author Response · Author response to Decision Letter 1]

28 Jun 2020

Manuscript ID: PONE-D-20-10365R1

Prevalence and factors associated with anemia among women of reproductive age in seven South and Southeast Asian Countries: evidence from nationally representative surveys

Dear Editor, 

Thank you very much for giving us an opportunity to submit a revised version of the manuscript entitled "Prevalence and factors associated with anemia among women of reproductive age in seven South and Southeast Asian Countries: evidence from nationally representative surveys" to reputed Journal PLOS ONE. We have revised the paper in response to the extensive and insightful reviewers' comments. We appreciate the time and efforts that you and the reviewers have dedicated to providing your constructive feedback on our manuscript. We believe the revisions have substantially improved our manuscript. 

We have highlighted the changes within the manuscript where necessary. We hope that the manuscript is now suitable for publication. We look forward to hearing a positive response from you.

Here is a point-by-point response to the academic editor and reviewers' comments and concerns. 

Reviewers' comments:

Reviewer #1

Comment: I believe that almost all comments were answered properly, which made the article clearer. Therefore, it can be published.

Response: Thank you very much for your constructive comments and suggestions. 

Comments: I suggest that in reference number 14 the name of the journal BMC Women’s Health be included, as this suggestion has not been heeded. However, after this correction, there is no need to return the article to the reviewers for further evaluation.

Response: Thank you very much for your valuable suggestion. We have included the name of the journal "BMC Women's Health" in the reference number 14 in the revised version of the manuscript. 

Reviewer #2:

Comments: Line 28-30: (Bangladesh DHS 2011, Cambodia DHS 2014, India NFHS 2016, Maldives DHS 2016, Myanmar DHS 2015, Nepal DHS 2016, Timor-Leste 2015)...

Change to: (Bangladesh, Cambodia, India, Maldives, Myanmar, Nepal and Timor-Leste)...

Response: We thank the reviewer. We have revised the sentence accordingly in the revised version of the manuscript, line 28-29.

Comments: Line 33: The combined prevalence of anemia among women of reproductive age in the seven selected South and Southeast Asian countries was…

Change to: The combined prevalence of anemia was…

Response: Thank you for your suggestion. We have revised the sentence accordingly with "The combined prevalence of anemia was 52.5%, ranged from 22.7% in Timor-Leste to 63.% in the Maldives". in revised manuscript line 32-33.

Comments: Line 66: …conducted by Balarajan Y., et al (2011) reported that […] urban or rural settings [6].

Change to: …conducted by Balarajan et al. [6] reported that [...] urban or rural settings.

Response: Thank you for pointing this out. We have revised the sentence with " A pooled analysis conducted by Balarajan et al [1] reported that the risk of anemia among women living in the lowest wealth quintile, with no education, and also differed by urban or rural settings". in revised manuscript line 65-66

Comments Line 113: Therefore, this study aims to identify […]: Start in a new paragraph.

Response: We have made a new paragraph of the sentence "Therefore, this study aims to identify the prevalence and factors associated with anemia among WRA in seven selected SSEA countries" in the revised manuscript line 112-113.

Comments: Line 143: […] anemia was defined […]: Put a comma between g/dL and respectively: (g/dL, respectively)

Response: Thank you for your comments. We have put a comma between g/dL and respectively with "According to the WHO, for non-pregnant and pregnant women aged 15-49 years, any form of anemia was defined as hemoglobin concentration <12.0 g/dL, and 11 g/dL respectively" in the revised manuscript line 141.

Comments: Line 144: Hemoglobin level was assessed using capillary blood and the HemoCue rapid testing technique […]

Wouldn't that be better? Hemoglobin level in capillary blood was assessed using the HemoCue rapid testing technique […]

Response: Thank you for your valuable suggestions. We have revised the sentences with "Hemoglobin level in capillary blood was assessed using HemoCue rapid testing technique in all seven South and Southeast Asian countries. For further analysis of the outcome variable, the categories of anemia were further dichotomized as anemic and not anemic". in the revised manuscript line 142-143.

Comments: Line 173: […] than one year. ANC visits during […]. In Portuguese, does not start sentence with abbreviations or numbers. I am not sure if there is this rule in English (I am not an English speaker).

Response: Thank you for pointing out this. We agree with you. We have replaced “ANC” with "antenatal checkup" in the revised manuscript line 170.

Comments: Line 206: Fig 1. Prevalence of anemia […]: Place after the next paragraph, in which fig. 1 is referred to for the first time.

Response: Thank you for your suggestion. We have moved the Fig 1 legend after line 211 in the consequent paragraph and text reference of Fig 1 has been also provided in the line 205. 

Comments: Line 329: […]. Yang F et al. (2018) also reported […] middle-income countries [38].

Change to: […]. Yang et al. [38] also reported […] middle-income countries.

Response: We have changed the sentence with “Yang et al. [2] also reported that the prevalence of anemia was observed higher among those individuals with low socioeconomic status from low and middle-income countries." accordingly in the revised manuscript, line 317.

Comments: Other similar cases exist throughout the text and should be corrected.

Response: We thank the reviewer for constructive comments. We have corrected it wherever necessary throughout the revised manuscript. 

We look forward to hearing a positive response from you soon.

Thank you. 

Sincerely

Mr. Dev Ram Sunuwar, Dietician/Researcher

Armed Police Force Hospital, Kathmandu, Nepal

Email: devramsunuwar@gmail.com

References 

1. Balarajan Y, Ramakrishnan U, Özaltin E, Shankar AH, Subramanian S V. Anaemia in low-income and middle-income countries. Lancet. 2011;378: 2123–2135. doi:10.1016/S0140-6736(10)62304-5

2. Yang F, Liu X, Zha P. Trends in Socioeconomic Inequalities and Prevalence of Anemia Among Children and Nonpregnant Women in Low- and Middle-Income Countries. JAMA Netw open. 2018;1: e182899. doi:10.1001/jamanetworkopen.2018.2899

---

## [Editor Report · Decision Letter 2]

1 Jul 2020

PONE-D-20-10365R2

Prevalence and factors associated with anemia among women of reproductive age in seven South and Southeast Asian countries: evidence from nationally representative surveys

PLOS ONE

Dear Dr. Sunuwar,

Thank you for submitting your manuscript to PLOS ONE. After careful consideration, we feel that it has merit but does not fully meet PLOS ONE’s publication criteria as it currently stands. Therefore, we invite you to submit a revised version of the manuscript that addresses the points raised during the review process.

The authors have made all changes requested by the reviewers. However, I have noted the use of the word "multivariable" instead of "multiple" regression models. Thus, please replace "multivariable" by "multiple" in the text and Tables when referring to multiple regression analyses. 

We look forward to receiving your revised manuscript.

Kind regards,

Marly A. Cardoso, Ph.D.

Academic Editor

PLOS ONE

---

## [Author Response · Author response to Decision Letter 2]

2 Jul 2020

Manuscript ID: PONE-D-20-10365R2

Prevalence and factors associated with anemia among women of reproductive age in seven South and Southeast Asian Countries: evidence from nationally representative surveys

Dear Editor,

We would like to thank you for allowing us to submit a revised copy of our manuscript entitled "Prevalence and factors associated with anemia among women of reproductive age in seven South and Southeast Asian Countries: evidence from nationally representative surveys" to reputed Journal PLOS ONE. We appreciate the time and effort that you and the reviewers have dedicated to providing your valuable feedback on our manuscript. We are grateful to the reviewers for their insightful comments on our papers. We believe the revisions have substantially improved our manuscript. 

We have replaced the word "multivariable regression model" with "multiple logistic regression model" in the text and Tables when referring to multiple regression analyses. We hope that the manuscript is now suitable for publication. 

We look forward to hearing a positive response from you.

Thank you. 

Sincerely

Mr. Dev Ram Sunuwar, Dietician/Researcher

Armed Police Force Hospital, Kathmandu, Nepal

Email: devramsunuwar@gmail.com

Here is a response to the academic editor's comments and concerns. 

Comment: The authors have made all changes requested by the reviewers. However, I have noted the use of the word "multivariable" instead of "multiple" regression models. Thus, please replace "multivariable" by "multiple" in the text and Tables when referring to multiple regression analyses.

Response: Thank you very much for your constructive comments and suggestions. We have replaced the word "multivariable regression model" with "multiple logistic regression model" in the text and Tables when referring to multiple regression analyses within the revised version of the manuscript.

---

## [Editor Report · Decision Letter 3]

8 Jul 2020

Prevalence and factors associated with anemia among women of reproductive age in seven South and Southeast Asian countries: evidence from nationally representative surveys

PONE-D-20-10365R3

Dear Dr. Sunuwar,

We’re pleased to inform you that your manuscript has been judged scientifically suitable for publication and will be formally accepted for publication once it meets all outstanding technical requirements.

Kind regards,

Marly A. Cardoso, Ph.D.

Academic Editor

PLOS ONE
---

## [Editor Report · Acceptance letter]

17 Jul 2020

PONE-D-20-10365R3 

Prevalence and factors associated with anemia among women of reproductive age in seven South and Southeast Asian countries: evidence from nationally representative surveys 

Dear Dr. Sunuwar:

I'm pleased to inform you that your manuscript has been deemed suitable for publication in PLOS ONE. Congratulations! Your manuscript is now with our production department. 

Kind regards, 

on behalf of

Dr. Marly A. Cardoso 

Academic Editor

PLOS ONE